# Resolution of Inflammation in Acute Graft-Versus-Host-Disease: Advances and Perspectives

**DOI:** 10.3390/biom12010075

**Published:** 2022-01-05

**Authors:** Layara Roberta Ferreira Duarte, Vanessa Pinho, Barbara Maximino Rezende, Mauro Martins Teixeira

**Affiliations:** 1Departamento de Morfologia, Instituto de Ciências Biológicas, Universidade Federal de Minas Gerais, Belo Horizonte 31270-901, Brazil; layararobertafduarte@gmail.com (L.R.F.D.); vpinhos@gmail.com (V.P.); 2Departamento de Enfermagem Básica, Escola de Enfermagem, Universidade Federal de Minas Gerais, Belo Horizonte 30130-100, Brazil; barbaramaximino@ufmg.br; 3Departamento de Bioquímica e Imunologia, Instituto de Ciências Biológicas, and Center for Advanced and Innovative Therapies, Universidade Federal de Minas Gerais, Belo Horizonte 31270-901, Brazil

**Keywords:** resolution, inflammation, GVHD, pro-resolving mediators

## Abstract

Inflammation is an essential reaction of the immune system to infections and sterile tissue injury. However, uncontrolled or unresolved inflammation can cause tissue damage and contribute to the pathogenesis of various inflammatory diseases. Resolution of inflammation is driven by endogenous molecules, known as pro-resolving mediators, that contribute to dampening inflammatory responses, promoting the resolution of inflammation and the recovery of tissue homeostasis. These mediators have been shown to be useful to decrease inflammatory responses and tissue damage in various models of inflammatory diseases. Graft-versus-host disease (GVHD) is a major unwanted reaction following allogeneic hematopoietic stem cell transplantation (allo-HSCT) and is characterized by an exacerbated inflammatory response provoked by antigen disparities between transplant recipient and donor. There is no fully effective treatment or prophylaxis for GVHD. This review explores the effects of several pro-resolving mediators and discusses their potential use as novel therapies in the context of GVHD.

## 1. Introduction

Inflammation is a reaction of the immune system to various stimuli and is believed to protect the body against pathogens and to facilitate healing following sterile tissue injury. Nonetheless, uncontrolled or unresolved inflammation can result in chronic inflammation, tissue injury, and consequent organ failure. The lack of resolution of the inflammatory process is thought to contribute to the pathogenesis and outcome of several inflammatory diseases, including arthritis, diabetes, neurodegenerative diseases, cardiovascular diseases, and inflammatory bowel diseases [1,2,3,4]. Therefore, the timely induction of the resolution of inflammation is crucial to dampen the inflammatory response and promote some key steps to restore tissue/organ homeostasis [5,6]. As discussed below, there is now significant evidence to suggest that the resolution of inflammation is an active process driven by the so-called pro-resolving mediators.

Allogeneic hematopoietic stem cell transplantation (allo-HSCT) is the main treatment for many hematological diseases. Graft-versus-host disease (GVHD) is an unwanted reaction of the transplanted cells to the host and may limit the success of allo-HSCT. GVHD is only second to malignant relapse as the cause of death in allo-HSCT recipients [7] and occurs when recipient antigen presenting cells (APCs) interact with transplanted donor T lymphocytes that recognize the host as non-self and attack host tissues, leading to tissue inflammation and injury [7,8]. Current treatment or prophylaxis for GVHD is only modestly effective, with an increased risk of infections, primary disease relapse, and long-term adverse effects. Despite intense efforts, there have been no significant advances in efficient approaches to prevent and control the exacerbated inflammatory response related to GVHD [9,10]. Therefore, new therapies that act by promoting inflammation resolution seem to be a promising way to provide better control of several crucial features of the pathogenesis of GVHD.

## 2. Inflammation

The acute inflammatory response is characterized by a sequence of coordinated cellular and molecular events involving the participation of soluble mediators, such as cytokines, chemokines, complements, eicosanoids, vasoactive amines, and free radicals, and leukocytes, including macrophages, neutrophils, lymphocytes, dendritic cells, endothelial cells, mast cells, and fibroblasts [11]. In the early phase of acute inflammatory response, tissue resident cells release soluble mediators, which cause protein exudation and leukocyte influx from the blood into tissues [5,11]. These leukocytes may engulf and eliminate invading pathogens and/or tissue debris. If this acute process is limited with the elimination of the inciting stimulus, the removal of infiltration cells occurs, the edema subsides and tissue may return to their usual homeostasis, a process referred to as the resolution of inflammation. However, persistence of the stimulation and a lack of resolution of the inflammatory response may lead to chronic inflammation, excessive tissue injury and fibrosis.

## 3. Resolution of Inflammation

Resolution of inflammation is the term used to describe the events that occur between the peak of leukocyte accumulation in the inflamed tissues, the complete removal of effector leukocytes and return of tissue to homeostasis [5,11]. In the past, the resolution of the inflammatory response was thought to be a passive process of elimination of pro-inflammatory cells and mediators. Currently, it is known that inflammation prompts an active process that is dependent on the action of endogenous pro-resolving mediators, which drive the termination of inflammation and restoration of tissue function [5,6,12].

The first step for resolution to occur is likely to be the decrease in or elimination of the stimulus responsible for the inflammatory response. We previously revised a series of events that occur at the time inflammation starts to subside and resolution kicks in [3]. Several molecules mediate the effective transition from the proinflammatory phase to the onset and establishment of resolution. Such mediators cause a decrease in the local chemokine concentration, a reduction that is essential to inhibit further infiltration of neutrophils into the tissue. Then, pro- and anti-inflammatory networks help to turn on the resolution program, which is characterized by an active switch in the mediators that predominate in exudates, with increases in molecules such as pro-resolving lipid mediators (e.g., Lipoxin A4) and annexin-A1 (Anx-A1). These initial steps will decrease the influx of neutrophils, and induce their apoptosis and engulfment by tissue macrophages, a process referred to as efferocytosis. Efferocytic macrophages change their phenotype and themselves are capable of inducing further resolution of inflammation [12]. Indeed, the cells that drive the initial response, such as neutrophils, eosinophils, or lymphocytes, are cleared from the inflamed tissue by monocytes recruited to the inflamed sites that differentiate into macrophages and change their pro-inflammatory profiles, and are defined as either M1 (or classically activated) macrophages, with a protective phenotype, or as M2 (or alternatively activated) macrophages. M1 macrophages are induced by microbial products or cytokines such as IFN-γ and TNF-α and produce high levels of pro-inflammatory cytokines, such as TNF-α, IL1-β, IL-6, IL-12, and IL-23, promoting Th-1 responses. On the other hand, M2 are induced by IL-4 or IL-13 and act in the resolution of local inflammation, through the clearing of apoptotic cells and cellular debris, repairing and regenerating tissue damage [13]. They express high levels of mannose receptor (CD206) and arginase-1, and produce TGF-β, insulin-like growth factor 1 (IGF-1) and IL-10 [14]. Moreover, recent reports have indicated the existence of a hybrid phenotype of classically and alternatively activated macrophages. These macrophages are named resolution-phase macrophages (rMs). rMs exist in two distinct subtypes: CD11b^high^ and CD11b^low^. CD11b^high^ macrophages express low levels of M1 markers and high levels of the M2 marker arginase-1. These CD11b^high^ macrophages secrete medium levels of inflammatory cytokines and chemokines, including IL-10. The CD11b^low^ macrophages express even lower levels of M1 markers, and they do not express arginase-1. Moreover, these cells secrete very low levels of inflammatory cytokines and chemokines, as well as IL-10, but greater amounts of TGF-β. In this way, rMs acts in the resolution of inflammation, expressing higher levels of anti-inflammatory, anti-fibrotic and anti-oxidant proteins to limit tissue damage and fibrosis [12].Until recently, it was believed that the resolution was the end of the immune response related to an infection or injury, but Newson et al. (2014) [15] demonstrated that the resolution of inflammatory response links the innate and adaptive immune system, constituting a third phase after acute inflammation and resolution, called the post-resolution phase. In this phase, there is an influx of adaptive immune cells, reassembly of resident macrophages and dendritic cells, and the adaptive immunity is established to maintain immune tolerance to endogenous antigens [15].

### 3.1. Pro-Resolving Mediators

Pro-resolving mediators are essential to trigger the events associated with resolution. These molecules are divided into different groups based on their chemical structures. The most studied pro-resolving mediators include [5,6]:

#### 3.1.1. Lipids

During resolution of inflammation, essential polyunsaturated fatty acids (PUFAs) are released from the cell membranes, leading to the synthesis of PUFA-derived pro-resolving mediators, which results in the activation of the resolution program. These mediators are formed by biosynthetic pathways that convert the arachidonic acid (AA) to lipoxins (LXs), ω-3-derived eicosapentaenoic acid (EPA) to E-series resolvins, and docosahexaenoic acid (DHA) to D-series resolvins, protectins and maresins. Lipoxins are known to inhibit leukocyte activation and infiltration to inflammation sites and to stimulate the uptake and clearance of apoptotic cells by macrophages, while resolvins can directly counter-regulate the biosynthesis of pro-inflammatory eicosanoids and suppress the production of pro-inflammatory markers [16,17]. D-series resolvins can act as a potent regulator of both human and murine PMN, may cause reductions in leucocyte infiltration, and can also block Toll-like receptor-mediated activation on macrophages [18]. Protectins can also reduce PMN infiltration, as well as stimulating the phagocytosis of apoptotic cells via macrophages [6,18]. Finally, the macrophage-derived resolution mediators, maresins, can inhibit neutrophil recruitment and stimulate the macrophage efferocytosis of apoptotic cells. Maresin 1 also presents analgesic actions controlling local inflammation and associated inflammatory pain [19].

#### 3.1.2. Proteins

The annexin-A1 (Anx-A1) is a protein mediator that regulates the inflammatory response in many models of inflammation. Anx-A1 can inhibit leukocyte recruitment to inflammatory sites by modulating both proinflammatory mediators and the antiinflammatory cytokine IL-10. In addition, Anx-A1, promotes the resolution of inflammation by inducing neutrophil apoptosis and increasing efferocytosis by macrophages [4,6]. The melanocortins (MC) are peptides derived from the cleavage of pro-opiomelanocortin, and include adrenocorticotropic hormone (ACTH), α-melanocyte-stimulating hormone (α-MSH), β-MSH and γ-MSH. In terms of evidence, it has been shown that the binding of α-MSH to MC receptors leads to the down-regulation of various proinflammatory cytokines. In addition, ACTH, α-MSH and other MC agonists can activate MC receptors on macrophages to reduce pro-inflammatory cytokines and chemokines, and enhance anti-inflammatory mediator levels [4,6]. Another protein, chemerin, is a natural ligand of ChemR23 (chemerin receptor 23). The chemerin/chemR23 axis can may be involved in the enhancement of phagocytosis on macrophages and the modulation of neutrophil-driven inflammation by inhibiting integrin activation, thus preventing excessive neutrophil trafficking to inflammatory loci and subsequent activation [6,20]. Galectins, a type of protein, are members of a large family of lectins that are highly conserved throughout animal evolution and are involved in various stages of immunity and inflammation, from initiation through resolution. Some members of this family, such as galectin-3, contribute to the pro-inflammatory response, whereas others, such as galectin-1 (Gal-1) and galectin-9 (Gal-9), contribute primarily to the resolution of inflammation, via the induction of T cell apoptosis, the stimulation of phagocytic clearance and the favoring of the differentiation of “alternatively activated” macrophages. In addition, in dendritc cells, Gal-1 controls tolerogenic or immunogenic programs by promoting the differentiation of IL-10-producing regulatory T [6,21,22,23].

#### 3.1.3. Gaseous Mediators

Gaseous mediators are molecules that diffuse freely through cell membranes (have no specific receptors), and have extremely low molecular weights and very short half-lives [6]. The major representatives of these molecules are hydrogen sulphide (H2S), nitric oxide (NO) and carbon monoxide (CO). H_2_S contributes significantly to promoting the repair of tissue injury and the restoration of tissue function. NO, as a pro-resolving molecule, can regulate apoptosis of inflammatory cells, and CO exhibits cell and tissue protection through anti-apoptotic, anti-inflammatory and anti-proliferative effects [24].

#### 3.1.4. Purine

Adenosine is a purine nucleoside that plays a crucial part in the regulation of immune homeostasis and can be found in every cell of the human body [6,25]. This molecule is an important anti-inflammatory agent and contributes to the resolution of inflammation at later stages of the inflammatory response, both by downregulating macrophage activation and by promoting Th2-versus Th1-cell development [26,27,28].

#### 3.1.5. Neuromodulators

Netrin-1 and acetylcholine are neuromodulators associated with the resolution of inflammation that act via the vagus nerve-mediated reflex. These molecules can attenuate the release of pro-inflammatory cytokines, increase the expression of M2 macrophages and inhibit leucocyte migration [6].

## 4. GVHD Overview

GVHD is classified as acute or chronic. Acute GVHD is the main fatal complication during the first months after allogeneic hematopoietic stem cell transplantation, while chronic GVHD is responsible for a significant long-term fraction of the mortality, morbidity and reduced quality of life of patients. Chronic GVHD (cGVHD) is a multiorgan pathology, and has autoimmune and fibrotic features. cGVHD can affect multiple organs including the skin (sclerosis), eyes (xerophthalmia), vagina, esophagus, liver, and lung (bronchiolitis obliterans), as well as causing serositis (including pericardial or pleural effusions), and, in rare cases, affecting the kidneys (nephrotic syndrome) [29]. Approximately 70% of patients that receive non-sibling marrow grafts develop chronic GVHD after day 100. Compared with the advances in our understanding of acute GVHD, the pathophysiology of chronic GVHD remains poorly defined [30]. Acute GVHD (aGVHD) usually manifests within the first 100 days after transplantation. About 50% of patients who receive allo-HSCT develop aGVHD, which can cause dysfunction of the gut, liver, skin and hematopoietic organs. The typical signs include maculopapular rash (skin), hyperbilirubinemia with jaundice, nausea, vomiting, watery and bloody diarrhea, and crampy abdominal pain. The only approved front-line therapy for both acute and chronic GVHD is systemic steroids [7,31,32,33]. The broad activity of steroids, including the induction of T-cell apoptosis, the suppression of macrophage activation, and the suppression of pro-inflammatory cytokine release, explains why these drugs are still the first-line treatment of both aGVHD and cGVHD. However, the administration of high doses and/or prolonged use of these drugs can cause serious side effects, such as an increase in the rate of opportunistic infections, diabetes, myopathy and osteonecrosis, which are factors that increase the morbidity and mortality associated with GVHD. Approximately 50% of steroid-treated patients do not improve, developing steroid-refractory aGVHD, and for them, there is no good long-term prognosis, with a survival rate of only 5 to 30% [32,34,35,36].

The pathophysiology of aGVHD is a complex process that can be divided into three phases. Initially, pathogen-associated molecular patterns (PAMPs), such as bacterial lipopolysaccharide (LPS), and damage-associated molecular patterns (DAMPs), such as adenosine triphosphate, are released from tissues, especially the intestine, in response to the chemotherapy or radiotherapy regimen (conditioning regimen). DAMPs and PAMPs are recognized by innate immune receptors, including Toll-like receptors (TLR). This interaction leads to the release of pro-inflammatory cytokines (“cytokine storm”), such as TNF-α, IL-1β and IL-6, which in turn, activate host antigen-presenting cells [32,37]. In the second phase, the interaction of donor T cells with activated APCs expressing MHC and minor host histocompatibility antigens leads to the activation and expansion of T cells [31]. In the third phase, named the effector phase, activated donor T cells and monocytes migrate to aGVHD target organs (skin, liver, spleen and intestine) and stimulate the recruitment of other effector cells, such as cytotoxic T cells and natural killer (NK) cells. These effector cells cause damage through direct cytotoxicity or by releasing large amounts of pro-inflammatory cytokines and chemokines, such as TNF-α, IL-1β, IL-2, IL-12, IL-17, IFN-γ, CCL2, CCL3, CCL4 and CCL5, which aggravate aGVHD and often cause death [31,38,39].

In patients undergoing allo-HSCT for the treatment of leukemia, it is important to mention that alloreactive donor T cells not only contribute to aGVHD but are also essential to attack host malignant cells, producing a beneficial effect known as graft-versus-leukemia (GVL). Although the depletion of T cells from the donor sample prior to allogeneic transplantation can prevent or reduce aGVHD, this also impairs the GVL response, with a consequent increase in relapse rates of the underlying disease [38]. Therefore, any effective treatment must be able to reduce aGVHD damage without compromising GVL activity.

In this review, we will primarily discuss the relevance of pro-resolving mediators studied in the context of acute GVHD.

## 5. Resolution of Acute GVHD: What Do We Know?

The possibility of using pro-resolving mediators to treat inflammatory diseases has led to a concept known as ‘resolution pharmacology, based on a new strategy to switch off complicated and often chronic inflammatory diseases [40]. In this context, a few studies have evaluated the relevance of pro-resolving agents in animal models of aGVHD, as summarized in Table 1.

Galectin-1 treatment was shown to reduce inflammatory infiltrates in the colon and liver, decrease IFN-γ on splenocytes and block host alloreactivity in a murine model of aGVHD. Gal-1 also improved normal splenic architecture after blood marrow transplantation and decreased mortality [41]. Sakai et al. (2011) [43] further demonstrated that Gal-9 inhibited the mixed lymphocyte reaction by inducing lymphocyte apoptosis, prevented the progression of aGVHD by reducing intestinal and liver injury, and decreased levels of TNF-α and IL-17. Moreover, Veenstra and colleagues (2013) [44] observed that Gal-9 transgenic mice (mice with high Gal-9 expression, especially in the small intestine, liver and lung) had a significant decrease in lethality rate compared to WT recipients.

Myeloid-derived suppressor cells (MSDCs) have various beneficial functions in the context of aGVHD. For example, they have immunosuppressive functions in relation to lymphocyte priming and expansion, suppress the effects of alloreactive T cells, induce the reprograming of Tregs and macrophages to an M2-like phenotype, accelerate tissue repair and downregulate inflammation by shifting the inflammatory environment with alterations in the production and types of cytokines and chemokines [59,60]. Infusion of MSDCs was found to enhance allogenic hematopoietic engraftment and prevent GVHD [42]. Yin et al. (2020) [42] observed that treatment with Gal-9 prior to transplantation enhanced the frequency of MDSC in the bone marrow, spleen and peripheral blood. Moreover, both Gal-9 treatment before transplant or transplant with MDSCs pre-treated with Gal-9 resulted in less damage to target organs, which was related to an increase in MDSC frequency and a suppression of T cell proliferation. However, in patients with aGVHD, these authors observed high concentrations of galectin-9 and an accumulation of MDSCs, which could be explained as an attempt to counterbalance the intense inflammatory response related to the disease. Thus, the participation of this molecule in the pathophysiology of GVHD is still unclear and needs to be further explored.

Another protein, ***Chemerin***, has attracted particular attention regarding multiple roles related to the control of inflammation [20]. In murine aGVHD, Vinci and colleagues (2015) [45] observed that mice that received ChemR23-deficient cells developed more severe aGVHD with significantly increased weight loss, more severe diarrhea and a higher mortality rate. The absence of the chemerin receptor also increased neutrophil frequency in the intestine and caused intense intestinal damage, characterized by crypt hyperplasia and atrophy, epithelium apoptosis and colitis. To the best of our knowledge, the effects of chemerin administration as a therapy for aGVHD has not been evaluated yet.

The conversion of arachidonic acid to bioactive lipid mediators through the lipoxygenase pathway may lead to the formation of leukotrienes and lipoxins (LX). At sites of inflammation, there is an inverse relationship between LX levels and leukotriene. Whereas leukotrienes are potent pro-inflammatory mediators that may enhance cytokine and free radical production by leukocytes [61], lipoxins, as presented above, are important mediators of resolution. [17]. Therefore, an imbalance in lipoxin–leukotriene homeostasis may be a key factor in the pathogenesis of aGVHD. Clinically, one may enhance the production of lipoxins and other pro-resolving lipids by offering eicosapentaenoic acid (EPA) and ω3-PUFAs in the diet. Takatsuka and colleagues (2001) [46] showed that treatment with EPA significantly decreased cytokine levels in the blood and reduced disease mortality in aGVHD patients. Preliminary data from Cuvelier et al. (2013) [47] also showed that dietary ω3-PUFAs, aspirin and aspirin-triggered lipoxins could modestly, but statistically significantly, improve survival and delay the onset of lethal aGVHD in mice, but the mechanisms related to these results have not been explored. The effects of the direct administration of pro-resolving lipid molecules in models of aGVHD remain to be determined.

Another pro-resolving mediator that has been extensively investigated in aGVHD is adenosine. In aGVHD, studies in murine models have already shown that treatment with adenosine agonists can decrease the incidence and severity of the disease, as well as improving survival. This occurs by reducing circulating levels of pro-inflammatory cytokines and chemokines, including IFN-γ, IL-6 and CCL2, increasing levels of IL-10 in the serum and suppressing the immune response in GVHD target tissues by increasing the number of Tregs in the spleen, peripheral blood, skin and colon. In addition, in vitro, treatment with adenosine and their agonists reduced alloreactive T lymphocytes in HLA-mismatched allogeneic co-cultures. Inhibition of CD39/adenosine signals in vitro also abolished the effect of human mesenchymal stem cells (MSCs, adult stem cells with immunoregulatory function) in reducing PBMC and T cell proliferation. In vivo, the inhibition of this pathway increases the aGVHD score and decreases survival. Similarly, Ni et al. (2019) [51] demonstrated that pretreatment of MSCs with the CD39 inhibitor reduces the protective effect of these cells in controlling the number of CD4^+^IL-17^+^ and CD4^+^ IFN-γ^+^ T cells in the spleen [48,49,50,51,52,53]. Thus, despite being considered a pro-resolving agent, the anti-inflammatory effects of adenosine have been explored more in the context of aGVHD than its pro-resolving actions.

Gaseous mediators also act as pro-resolving mediators in different inflammatory diseases, but little is known about their role in aGVHD. So far, only nitric oxide (NO) has been studied in the disease and the evidence for the role of NO as a pro-resolving molecule is small [5]. Ren et al. (2008) [55] demonstrated, in a murine model of aGVHD, that the immunosuppression exerted by MSCs occurs through the production of NO after stimulation with pro-inflammatory cytokines. MSCs in culture with splenocytes fail to inhibit splenocyte proliferation in the presence of a selective inhibitor of iNOS (NMMA). Furthermore, aGVHD mice treated with *iNOS*^−/−^ MSCs showed increased infiltration of lymphocytes in the liver, lungs and skin as well as a decrease in survival rate. Moreover, the suppression of NO production in a murine model of aGVHD was associated with enhanced weight loss early post-transplant and decreased overall survival, because NO inhibition impaired hematopoietic reconstitution [57]. Furthermore, macrophage-produced NO resulted in a potent cytostatic effect that inhibited tumor cell proliferation in mice, but despite this cytostatic effect, consistently high levels of NO were found in mice with lethal aGVHD and transient levels were found in mice with nonlethal aGVHD [54]. Thus, this molecule appears to play a dual role in inflammation. Plasma levels of NO are related with the occurrence of moderate to severe aGVHD in hematopoietic stem cell transplant recipients. Several studies have also demonstrated that there is an increase in NO levels in both aGVHD mice and in patients with aGVHD [56,57,58]. Hoffman and colleagues (1997) [58] showed that treatment with aminoguanidine (an inhibitor of NO synthesis) decreased serum NO levels and improved murine aGVHD pathology, significantly reducing lethality.

As described above, the reprogramming of macrophages from classically activated to alternatively activated and stimulating the production of anti-inflammatory mediators for tissue repair and regeneration are crucial steps for the resolution of inflammation. Many studies have shown that reducing the number of M1 cells and increasing the number of M2 cells in mice clearly improves survival and reduces the severity of GVHD. The infusion of anti-inflammatory M2 alleviates aGVHD by downregulating the activity of T cells, which is characterized by decreased proliferation and decreased percentages of Th1 and Th17. Moreover, Hanaki et al. (2021) [62] demonstrated that M2 macrophages improve aGVHD by inhibiting the expansion of alloreactive T lymphocytes in a cell contact-dependent manner via the programmed death-ligand 1 and 2 (PD-L1/PD-L2) pathways. Other studies in the murine model of aGVHD also showed that M2 suppresses T cell responses by producing arginase 1 and IL-10, which suppress T cell infiltration. Similarly, macrophage polarization is also involved in the pathogenesis of aGVHD in humans. Some studies have shown that macrophages recruitment is one of hallmarks in the initiation of aGVHD and a higher ratio of M1/M2 correlates with a higher incidence of grade 2–4 aGVHD. Patients with aGVHD also present macrophages polarized towards pro-inflammatory M1. On the other hand, the infusion of M2 macrophages derived from donor BM attenuated the severity of aGVHD and prolonged survival after HSCT [62,63,64,65,66].

Regulatory T cells (Tregs) stimulate the production of anti-inflammatory mediators to control the inflammatory response. Tregs are critical for immune homeostasis and tolerance induction after HSCT. The role of these cells in resolution is emerging as a relevant topic, since they suppress cytotoxic cells, inhibit the secretion of pro-inflammatory cytokines by cytotoxic cells and secrete anti-inflammatory cytokines [67,68]. Tregs that express FOXP3 have been shown to be essential for maintaining tolerance and may suppress aGVHD in patients. A low CD4^+^FOXP3^+^ T-cell count early after HSCT was associated with an increased risk of aGVHD, and the ratio of Tregs to effector T cells was significantly reduced in patients with graft-versus-host disease. In addition, several studies have already showed, in murine models of aGVHD, that Tregs suppress the priming, expansion and/or function of effector T cells and that this protection is associated with IL-10 production [69,70,71,72,73].

Many intracellular signaling pathways involved in maintaining leukocyte survival, apoptotic death and the clearance of effector leukocytes have also been widely explored in the resolution of inflammation, including the phosphoinositide-3-kinase (PI3K) pathway [74]. Numerous studies have already shown that PI3Ks are essential in many cellular processes, including vesicular trafficking, cytoskeletal organization, cell survival, proliferation and growth. PI3K activation in T cells promotes survival, modulates differentiation and controls the acquisition of effector and memory phenotypes. Herrero-Sánchez et al. (2016) [75] demonstrated in human T cell cultures that selective PI3K inhibitors decrease T cell proliferation, inhibit T cell responses and induce tolerance of alloreactive T cells while preserving the immune response against cytomegalovirus. Using a model of GVHD in mice, these authors showed that a selective PI3K inhibitor (BEZ235) could have a potential therapeutic benefit in the prophylaxis of aGVHD. Treatment with BEZ235 significantly improved mice survival and mitigated the aGVHD-associated inflammatory response. Additionally, Xing and colleagues [76], suggested that PI3K expression is increased in patients with GVHD compared to patients without GVHD, and is correlated with different stages of the disease. Similarly, our group found that PI3K controls leukocyte recruitment, tissue injury and lethality in aGVHD. Although we did not directly investigate the role of this molecule in leukocyte survival, there was a decrease in CD8^+^ and CD4^+^ T cells in PI3K-deficient mice subjected to aGVHD, suggesting that this pathway may also be relevant for maintaining leukocyte survival and extending inflammation related to GVHD [77].

## 6. Experience from IBD

The experience with models of inflammatory bowel disease (IBD) may be useful to understand the potential relevance and applicability of the use of pro-resolving molecules in the context of aGVHD. Indeed, IBD shares certain similar pathophysiological mechanisms and multiple key features with aGVHD, including intestinal tissue damage and loss of intestinal barrier function. However, they do differ in significant ways as allogeneic HSCT is primarily an autoaggressive condition with switching of fundamental antigen-presenting mechanisms in HLA incompatibility. For IBDs, the role of some pro-resolving mediators was explored [78], suggesting that pro-resolving molecules may also be beneficial in aGVHD. In murine models of IBD, D-series resolvins reduced neutrophil infiltration and inhibited the release and expression of cytokines, chemokines and adhesion molecules [79,80]. In Dextran Sulphate of Sodium (DSS)-induced or 2,4,6-trinitrobenzene sulfonic acid (TNBS)-induced colitis, treatment with E-series resolvin (RvE1) suppressed the pro-inflammatory responses of macrophages, inhibited the release of pro-inflammatory cytokines, increased survival rates, improved histological scores and decreased leukocyte infiltration [81,82,83]. It has already been shown that protectin D1 treatment decreased leukocyte-endothelial interaction, attenuated neutrophil migration during inflammation and reduced mucosal TNF-α, IL-1β and IL-6, thereby leading to improvements in a IBD murine model. In addition, some studies have demonstrated that maresin treatment significantly reduced inflammatory cytokines production, restored body weight, improved the expression of tight junction proteins, reduced the infiltration of neutrophil and macrophages and induced a switch to the M2 macrophage phenotype in a colitis rodent model [80,84,85,86]. In the colitis model, treatment with annexin-A1 regulated the microbial community, decreased leukocyte migration and induced neutrophil apoptosis. Additionally, decreased expression of Anx-A1 was detected in the colonic mucosa of IBD patients, which correlated with increased levels of TNF-α transcripts and gut mucosal epithelial barrier disruption [87,88]. A few studies have also demonstrated that the absence of melanocortin-1 receptors resulted in higher weight loss and marked histological changes in the gut of mice with colitis. Furthermore, treatment with melanocortin-derived tripeptide reduced the activity of inflammatory infiltrate and myeloperoxidase in colon tissue [89,90,91]. Gaseous mediators also had a protective effect in terms of the intestinal damage related to colitis in mice [92,93,94,95,96,97,98]. Altogether, these results clearly demonstrate the gut-protective effects of pro-resolving mediators. Whether these molecules will have similar protective effects in models of aGVHD is not entirely known but further experimentation is clearly warranted.

## 7. Concluding Remarks and Future Directions

As discussed in this review, some mediators with known pro-resolving activity have been studied in the context of aGVHD and shown to reduce inflammation and damage to target organs in addition to overall mortality. Figure 1 and Figure 2 summarize the effects of the tested mediators in vivo and in vitro, in an attempt to identify potential site of action of these mediators. It is of note that most molecules tested in animal models of aGVHD to date are not the most well-defined pro-resolving molecules (see the review published by Sugimoto et al., 2019 [5] for a list of well-defined molecules) and, hence, have other effects that could account for their beneficial effects. The published studies have failed to convincingly show that pro-resolving activity (i.e., induction of apoptosis, efferocytosis and change in macrophage phenotype) are more relevant than anti-inflammatory activities (i.e., inhibition of production of cytokines/chemokines and inhibition of leukocyte influx and activation) for the observed protective effects. Indeed, most studies have focused primarily on the anti-inflammatory actions. Anti-inflammatory mediators act primarily by inhibiting leukocyte activation recruitment, and by reducing vascular permeability and endothelial cell activation. On the other hand, pro-resolving mediators induce leukocyte apoptosis and efferocytosis, and change the macrophage phenotype. These actions alter the progression of the inflammatory process and eventually lead to the activation of different signaling cascades to end the inflammatory response and promote subsequent tissue resolution and repair.

Future studies will, therefore, need to investigate the effects of mediators with more defined pro-resolving activity, such as lipoxins, annexin-A1, melanocortin and angiotensin 1–7 (or mimetics of these molecules), and show that protection is actually associated with pro-resolving activities. The studies discussed here have been carried out in the context of IBD and suggest that resolution pharmacology may also be beneficial for aGVHD, a tenet that clearly deserves further experimentation. The future will show whether pro-resolving molecules will be an efficient treatment option for patients who do not respond to classic anti-inflammatory corticosteroid-based first-line therapy, alone or in addition to the well-established first line. Altogether, these studies may open interesting new possibilities for testing in the context of aGVHD.

## Figures and Tables

**Figure 1 biomolecules-12-00075-f001:**
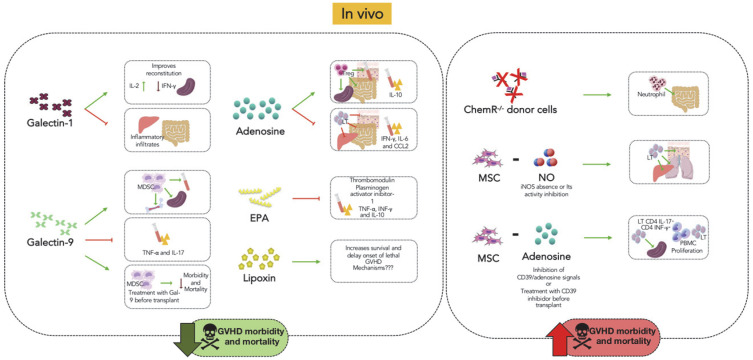
Summary of the in vivo effects of mediators with known pro-resolving actions in models of GVHD. Galectins, adenosine, EPA, lipoxins, chemerin and NO have already been studied at GVHD. In general, all these pro-resolving mediators contribute to the reduction in the inflammatory response in target organs, which in turn increases survival and decreases the clinical signs of the disease. The suggested mechanisms of control of the inflammatory response of each molecule are shown in the figure. It is noteworthy that the dual role of NO in contributing to the enhancement of the inflammatory response in GVHD was not represented here. MDSC: Myeloid-derived suppressor cells; EPA: eicosapentaenoic acid; T-reg: regulatory T cell; MSC: mesenchymal stem cells; ChemR^−/−^: chemerin receptor; NO: nitric oxide; PBMC: peripheral blood mononuclear cell.

**Figure 2 biomolecules-12-00075-f002:**
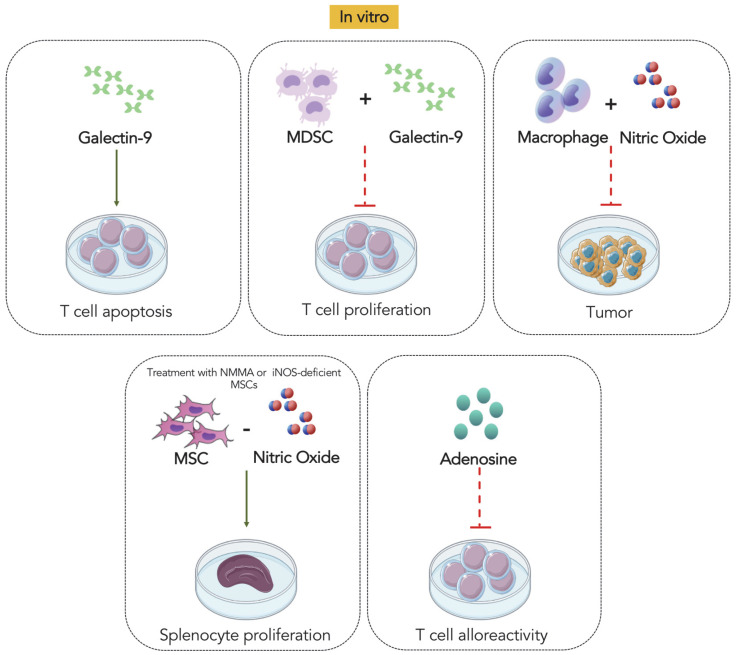
Summary of the in vitro actions of mediators with known pro-resolving activity. Galectin-9, adenosine and NO have demonstrated important actions in GVHD in vitro, including the apoptosis of lymphocytes T (by Gal-9), the reduction in T cell proliferation (by Gal-9-induced MDSCs) and alloreactivity (by Adenosine), the inhibition of tumor cells (via NO produced by peritoneal macrophages isolated from GVHD), and the reduction in splenocyte proliferation (via NO produced by MSCs). MDSC: Myeloid-derived suppressor cells; MSC: mesenchymal stem cells.

**Table 1 biomolecules-12-00075-t001:** Main actions of pro-resolving mediators in models of GVHD.

Pro-Resolving Mediators		
Mediator	Action	References
Galectin 1 (Gal-1)	In a murine model (treatment with Gal-1): Reduces disease mortality; reduces inflammatory infiltrates in colon and liver; improves reconstitution of normal splenic architecture following transplant; improves IL-2 and reduces IFN-γ on splenocytes; reduces host alloreactivity.	[41]
Galectin 9 (Gal-9)	In a murine model (treatment with Gal-9 before transplant): Increases MDSC frequencies in bone marrow, spleen and peripheral blood [42]; reduces T cell proliferation [29]; reduces TNF-α and IL-17 in peripheral blood [43]; decreased damage to intestine and liver [43]; reduces disease mortality and GVHD score [43]. In a murine model (mice with high gal-9 expression: “gal-9 transgenic mice”): Decrease in lethality rate compared to WT recipients [44]. In a murine model (MDSCs treated with Gal-9 before transplant): Reduces T cell proliferation; reduces disease mortality and GVHD score [42]. In HLA-mismatched allogeneic co-cultures: Induces T-cell apoptosis [43]. In cocultures of MSDCs and TCD3^+^: Gal-9-induced MDSCs suppressed T cell proliferation [42].	[43,44]
Chemerin	In a murine model (absence of the chemerin receptor, ChemR23^−/−^ in the donor cells): Causes more severe GVHD, increasing mortality and weight loss, associated with severe diarrhea; aggravates histopathological score in large intestine that presents crypt hyperplasia and atrophy, epithelium apoptosis and colitis; amplifies neutrophils infiltration in large intestine.	[45]
Eicosapentaenoic acid (EPA)	In GVHD patients (treatment with EPA): Reduces TNF-α, IFN and IL-10 in blood samples; reduces thrombomodulin and plasminogen activator inhibitor-1 in blood samples; reduces disease mortality.	[46]
Lipoxin	In a murine model (dietary ω3-PUFAs, aspirin, and aspirin-triggered lipoxins): Increases survival and delays the onset of lethal GVHD.	[47]
Adenosine	In a murine model (treatment with adenosine agonists): Reduces serum of IFN-γ, IL-6 and CCL2 [48]; increases serum IL-10 [48,49]; reduces CD3^+^ T cells in the liver, skin and colon [48]; increases Tregs on spleen [49], peripheral blood [49], colon [50] and skin [50]; reduces focal ulceration in skin and less edema in colon [48]; limits the severity of GVHD, inhibiting weight loss and improving survival [48,49]. In a murine model (MSCs pre-treatment with CD39 inhibitor): Reduces protective effects of MSCs associated with control of the number of TCD4^+^IL-17^+^ and TCD4^+^ IFN-γ^+^ in spleen [51]. In HLA-mismatched allogeneic co-cultures: Reduces alloreactivity of T lymphocytes [52]; In a humanized model (treatment with CD39 inhibitor): Inhibition of CD39/adenosine signals abolishes the effect of human MSCs on the reduction in the proliferation of PBMC and T cells, increases the GVHD score and decreases survival [53].	[48,49,50,51,52,53]
Nitric oxide (NO)	Beneficial effects: In a murine model/cellular culture: Peritoneal macrophages isolated from GVHD mice and primed with LPS in vitro, produced NO that promoted the cytostatic effect, inhibiting tumor cell proliferation [54]. In a murine model (transplant of *iNOS*^−/−^ MSCs): Inhibits protective effect of MSCs; survival and GVHD score were similar to untreated positive controls [55]. In a murine model (treatment with a selective inhibitor of iNOS activity—NMMA): increases lymphocytes infiltration in liver, lungs and skin [55]. In cocultures of MSCs and splenocytes: Treatment with NMMA or iNOS-deficient MSCs failed to inhibit splenocyte proliferation [55]; Harmful effects: In GVHD patients: Plasma levels of NO are related with the occurrence of moderate to severe aGVHD [56,57]. In a murine model: Treatment with aminoguanidine (an inhibitor of NO synthesis) decreased serum NO levels and ameliorated GVHD pathology, significantly reducing lethality [58].	[54,55,56,57,58]

## Data Availability

Not applicable.

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
