# Peer review of "Resolution of Inflammation in Acute Graft-Versus-Host-Disease: Advances and Perspectives"

_biomolecules, 2022, doi:10.3390/biom12010075_

Round 1
Reviewer 1 Report
This Review identifies a relevant point of view which needs to be addressed in future therapeutic developments for GVHD. It reads well and well structured.
I only have minor comments to refine the authors Statements and presentation of GVHD as clinical entity. Some wording regarding efficacy of GVHD prophylaxis and treatment is imbalanced. There are several options who definitely improved the reduction of GVHD rates over the recent decade, incl ATG/PTCY. The authors should be careful when using strong language in order to underline the necessity of this Review. They do not need to do that.
Also please go through the GVHD classification again, it is not just pre day 100 and post day 100 which classifies GVHD. Be more differentiated here or leave it.
Author Response
We are grateful for the comments on our manuscript entitled: “Resolution of Inflammation in Acute Graft-versus-host-disease: Advances and Perspectives”. We appreciate the reviewer’s comments and suggestions. Please find attached the modified manuscript and a list of actions taken in response to the reviewer's critiques.
Reviewer 1:
- This Review identifies a relevant point of view which needs to be addressed in future therapeutic developments for GVHD. It reads well and well structured. I only have minor comments to refine the authors Statements and presentation of GVHD as clinical entity. Some wording regarding efficacy of GVHD prophylaxis and treatment is imbalanced. There are several options who definitely improved the reduction of GVHD rates over the recent decade, incl ATG/PTCY. The authors should be careful when using strong language in order to underline the necessity of this Review. They do not need to do that. Also please go through the GVHD classification again, it is not just pre day 100 and post day 100 which classifies GVHD. Be more differentiated here or leave it.
We appreciate the suggestions made by the reviewer, and we have made a set of general changes throughout the manuscript to improve it. Also, we have modified the text in order to clarify the distinction between acute and chronic GVHD. Lines: 231-237.
Reviewer 2 Report
The review paper follows clear research aims. It concerns general pathways of immune inflammation and its recovery, with graft-versus-host disease (GVHD) taken as a particular example. Resolution of inflammation seems to be controlled by appropriate endogenous immune mediators. The authors support a concept of pro-resolving mediators which could be produced, e.g., by the resolution-phase macrophages. In principle, it could be applied to GVHD management. The general idea of the paper is quite rational.
Remarks:
Lane 108-111 (page 3): the list of pro-resolving mediators should be more strictly classified, referring the existing studies, with respect to the distinct inflammatory mechanisms, e.g., annexin, lipoxin, protectin, maresin etc. In further text, their effects are not referred.
Page 3. In the GVHD overview, the authors clearly discern acute and chronic GVHD, however from Lane 153-169 (page 4), the data seem to concern acute GVHD only, or they are similar for both GVHD types?
Page 4-5: Several potential protein and cellular factors are listed which promote GVHD resolution. Which of these factors are pertinent to acute GVHD, and what could act at chronic GVHD?
Page 7 (Section 6) concerns IBD pathology. This disorder seems to be of something other origin. Despite certain autoimmune component. By opposite, GVHD folloming allogeneic HSCT is, primarily, an autoaggressive condition, with switching of fundamental antigen-presenting mechanisms in HLA incompatibility. There is no need to directly compare these conditions, despite some lines of similarity. Hence, the IBD section could be sufficiently abridged.
The review paper follows clear research aims. It concerns general pathways of immune inflammation and its recovery, with graft-versus-host disease (GVHD) taken as a particular example. Resolution of inflammation seems to be controlled by appropriate immune mediators. The authors support a concept of pro-resolving mediators which could be produced, e.g., by the resolution-phase macrophages. In principle, it could be applied to GVHD management. The general idea of the paper is quite rational.
Remarks:
Lane 108-111 (page 3): the list of pro-resolving mediators should be more strictly classified, referring the existing studies, with respect to the distinct inflammatory mechanisms, e.g., annexin, lipoxin, protectin, maresin etc. In further text, their effects are not referred.
Page 3. In the GVHD overview, the authors clearly discern acute and chronic GVHD, however from Lane 153-169 (page 4), the data seem to concern acute GVHD only, or they are similar for both GVHD types?
Page 4-5: Several potential protein and cellular factors are listed which promote GVHD resolution. Which of these factors are pertinent to acute GVHD, and what could act at chronic GVHD?
Page 7 (Section 6) concerns IBD pathology. This disorder seems to be of something other origin. Despite certain autoimmune component. By opposite, GVHD following allogeneic HSCT is, primarily, an autoaggressive condition, with fundamental antigen-presenting mechanisms due to HLA incompatibility between donor and recipient. There is no need to directly compare these conditions, despite some lines of similarity. Hence, the IBD section could be sufficiently abridged.
Author Response
We are grateful for the comments on our manuscript entitled: “Resolution of Inflammation in Acute Graft-versus-host-disease: Advances and Perspectives”. We appreciate the reviewer’s comments and suggestions. Please find attached the modified manuscript and a list of actions taken in response to the reviewer's critiques.
Reviewer 2:
The review paper follows clear research aims. It concerns general pathways of immune inflammation and its recovery, with graft-versus-host disease (GVHD) taken as a particular example. Resolution of inflammation seems to be controlled by appropriate endogenous immune mediators. The authors support a concept of pro-resolving mediators which could be produced, e.g., by the resolution-phase macrophages. In principle, it could be applied to GVHD management. The general idea of the paper is quite rational.
Remarks:
Lane 108-111 (page 3): the list of pro-resolving mediators should be more strictly classified, referring the existing studies, with respect to the distinct inflammatory mechanisms, e.g., annexin, lipoxin, protectin, maresin etc. In further text, their effects are not referred.
We added this information to a subsection “Pro-resolving mediators”. Lines: 127-206.
Page 3. In the GVHD overview, the authors clearly discern acute and chronic GVHD, however from Lane 153-169 (page 4), the data seem to concern acute GVHD only, or they are similar for both GVHD types?
Page 4-5: Several potential protein and cellular factors are listed which promote GVHD resolution. Which of these factors are pertinent to acute GVHD, and what could act at chronic GVHD?
Information about cGVHD has been inserted for context, but the purpose of this review is focused on aGVHD. Therefore, we add the letter “a” before GVHD throughout the text to specify that we are only dealing with acute in the presented results. We also added the following sentence at the end of the section GVHD overview, lines 283-284: "In this review we will discuss primarily the relevance of pro-resolving mediators studied in the context of acute GVHD".
Page 7 (Section 6) concerns IBD pathology. This disorder seems to be of something other origin. Despite certain autoimmune component. By opposite, GVHD following allogeneic HSCT is, primarily, an autoaggressive condition, with switching of fundamental antigen-presenting mechanisms in HLA incompatibility. There is no need to directly compare these conditions, despite some lines of similarity. Hence, the IBD section could be sufficiently abridged.
In keeping with the suggestion of the reviewer, we shortened this paragraph and also highlighted differences between IBD and HSCT. Lines: 451-497.
Reviewer 3 Report
Major comments
- Line 135-136; This statement should be supported by a reference.
- Check with other articles in the journal and author’s instructions, but Table 1 according to my knowledge of MDPI journals, is that figures and tables should appear as close as possible to in-text citation. If so Table 1 needs to be moved up.
- Lines 179-182: This opening paragraph does not set the use of ‘resolution pharmacology’ in GVHD sufficiently. Either here or elsewhere in the manuscript, it needs to be clarified if ‘resolution pharmacology’ will be sufficient in treating GVHD or whether it is an additional treatment once the cause of the GVHD has been dealt with first. In this regard, for instance, I noted in a quick search a recent clinical trial using “subcutaneous injections of uhCG in addition to standard immunosuppression” (DOI: 10.1182/bloodadvances.2019001259)
- The text regarding Ref 46-51 does not align with Table and effects of adenosine may be directly on T cells, rather than via in MDSCs, in some instances. Additional text and separation of these treatments (e.g. MDSCs vs T cells) is required to align better with Table.
- Line 341: It is unclear if “different” refers to the various resolvins discussed below or different resolvins to those observed in GVHD. Given those listed it would seem the latter. Please clarify this in the text.
- Table 1: The use of capital letters after each “;” is grammatically incorrect. Change all instances to lower case.
- Figures 1 and 2: Figures would be clearer if all abbreviations were (re)explained in each legend.
- Figures 1 and 2: Bold font is used in Fig 1 but not Fig 2 – standardize font for consistency.
- Figure 2: “T cell alloreactive” should be “T cell alloreactivity” to align with other verbs used.
Further editing required (tip: some of the below can be identified and addressed using the ‘Find’ function in Word)
Title: Case in title is incorrect: “acute” should be changed to (=) “Acute” and “perspectives” = “Perspectives”.
Lines 34-35: “As it will discussed below” = “As discussed below”
Line 78: “Then Pro- and Anti-“ to “Then pro- and anti-“
Lines 85 & 343: “PMNs/PMN” = “neutrophils/neutrophil” to match other instances
Lines 87-90 onwards: The terminology regarding macrophages is unclear and not consistent throughout manuscript. I suggest changing lines 87-90 to “…as M1 (or classically activated) macrophages to a protective phenotype, referred to M2 (or alternatively activated) macrophage and/or resolution-phase macrophages (rMs). These rMs…” Then please check other instances throughout manuscript for consistency, especially but not limited to lines 284-302.
Line 92: “secret” = “secrete”
Line 95: “cytokines and chemokines, and IL-10” “cytokines and chemokines, including IL-10”
Line 116-117: Correct indent/alignment and some phrasing is redundant. I suggest “induction of prompt phagocytosis of apoptotic leukocyte by macrophages (efferocytosis);”
Line 135: Grammatically incorrect to start sentence with Arabic numeral. I suggest “Some 70%” or “Approximately 70%”
Line 150: Insistence use of “acute GVHD” – change to “aGVHD” and check throughout
Line 165: “Natural Killer” = “natural killer”
Line 202: Insistence use of “graft-versus-host disease” – change to “GVHD” and check throughout.
Lines 232 and 235: in “u3” the “u” should be omega symbol or “omega”
Line 253: “signals also in vitro” = “signals in vitro also”
Line 258: “TCD4+” is unconventional – change to “CD4+IL-17+ and CD4+IFN-γ+ T cells”.
Line 266” Delete “,” after Ref 52.
Line 295: “arginase 1 enzyme and” = “arginase 1 and”
Line 322: “T cell culture” = “T cell cultures”
Lines 344-345: Explain “DSS” and “TBNS” in full (without stating abbreviations) for readers outside the field of IBD
Lines 353 and 362: Unnecessary abbreviations (MaR1, MC1R) as not used again – remove according and check manuscript for any other unnecessary abbreviations
Line 364: “MPO” = “myeloperoxidase”
Lines 366-370: Subscript “2” in “H2S”
Line 397: “annexin-A1” = ANAX-1” and check all other abbreviations are correctly used
Table 1: “CheR23-/-,” runs over line – correct
Table 1: “in vitro” not italicised. Check all instances (and also for in vivo) throughout and correct accordingly
Author Response
We are grateful for the comments on our manuscript entitled: “Resolution of Inflammation in Acute Graft-versus-host-disease: Advances and Perspectives”. We appreciate the reviewer’s comments and suggestions. Please find attached the modified manuscript and a list of actions taken in response to the reviewer's critiques.
Reviewer 3:
Major comments
- Line 135-136; This statement should be supported by a reference.
- Check with other articles in the journal and author’s instructions, but Table 1 according to my knowledge of MDPI journals, is that figures and tables should appear as close as possible to in-text citation. If so Table 1 needs to be moved up.
These minor points were corrected in the text.
- Lines 179-182: This opening paragraph does not set the use of ‘resolution pharmacology’ in GVHD sufficiently. Either here or elsewhere in the manuscript, it needs to be clarified if ‘resolution pharmacology’ will be sufficient in treating GVHD or whether it is an additional treatment once the cause of the GVHD has been dealt with first. In this regard, for instance, I noted in a quick search a recent clinical trial using “subcutaneous injections of uhCG in addition to standard immunosuppression” (DOI: 10.1182/bloodadvances.2019001259).
This point was clarified in the section “Concluding remarks and Future directions”, lines 525-526.
- The text regarding Ref 46-51 does not align with Table and effects of adenosine may be directly on T cells, rather than via in MDSCs, in some instances. Additional text and separation of these treatments (e.g. MDSCs vs T cells) is required to align better with Table.
- Line 341: It is unclear if “different” refers to the various resolvins discussed below or different resolvins to those observed in GVHD. Given those listed it would seem the latter. Please clarify this in the text.
- Table 1: The use of capital letters after each “;” is grammatically incorrect. Change all instances to lower case.
- Figures 1 and 2: Figures would be clearer if all abbreviations were (re)explained in each legend.
- Figures 1 and 2: Bold font is used in Fig 1 but not Fig 2 – standardize font for consistency.
- Figure 2: “T cell alloreactive” should be “T cell alloreactivity” to align with other verbs used.
These minor points were corrected in the text.
Further editing required (tip: some of the below can be identified and addressed using the ‘Find’ function in Word)
Title: Case in title is incorrect: “acute” should be changed to (=) “Acute” and “perspectives” = “Perspectives”.
Lines 34-35: “As it will discussed below” = “As discussed below”
Line 78: “Then Pro- and Anti-“ to “Then pro- and anti-“
Lines 85 & 343: “PMNs/PMN” = “neutrophils/neutrophil” to match other instances
These minor points were corrected in the text.
Lines 87-90 onwards: The terminology regarding macrophages is unclear and not consistent throughout manuscript. I suggest changing lines 87-90 to “…as M1 (or classically activated) macrophages to a protective phenotype, referred to M2 (or alternatively activated) macrophage and/or resolution-phase macrophages (rMs). These rMs…” Then please check other instances throughout manuscript for consistency, especially but not limited to lines 284-302.
This point was clarified in the lines 51-110.
Line 92: “secret” = “secrete”
Line 95: “cytokines and chemokines, and IL-10” “cytokines and chemokines, including IL-10”
Line 116-117: Correct indent/alignment and some phrasing is redundant. I suggest “induction of prompt phagocytosis of apoptotic leukocyte by macrophages (efferocytosis);”
Line 135: Grammatically incorrect to start sentence with Arabic numeral. I suggest “Some 70%” or “Approximately 70%”
Line 150: Insistence use of “acute GVHD” – change to “aGVHD” and check throughout
Line 165: “Natural Killer” = “natural killer”
Line 202: Insistence use of “graft-versus-host disease” – change to “GVHD” and check throughout.
Lines 232 and 235: in “u3” the “u” should be omega symbol or “omega”
Line 253: “signals also in vitro” = “signals in vitro also”
Line 258: “TCD4+” is unconventional – change to “CD4+IL-17+ and CD4+IFN-γ+ T cells”.
Line 266” Delete “,” after Ref 52.
Line 295: “arginase 1 enzyme and” = “arginase 1 and”
Line 322: “T cell culture” = “T cell cultures”
Lines 344-345: Explain “DSS” and “TBNS” in full (without stating abbreviations) for readers outside the field of IBD
Lines 353 and 362: Unnecessary abbreviations (MaR1, MC1R) as not used again – remove according and check manuscript for any other unnecessary abbreviations
Line 364: “MPO” = “myeloperoxidase”
Lines 366-370: Subscript “2” in “H2S”
Line 397: “annexin-A1” = ANAX-1” and check all other abbreviations are correctly used
Table 1: “CheR23-/-,” runs over line – correct
Table 1: “in vitro” not italicised. Check all instances (and also for in vivo) throughout and correct accordingly
These minor points were corrected in the text.
Reviewer 4 Report
The review is short and sweet. Layara et al. provided a good summary of inflammation in aGVHD. I endorse the publication of this review without any further modifications.
Author Response
We are grateful for the comments on our manuscript entitled: “Resolution of Inflammation in Acute Graft-versus-host-disease: Advances and Perspectives”. We appreciate the reviewer’s comments and suggestions. Please find attached the modified manuscript and a list of actions taken in response to the reviewer's critiques.
Reviewer 4
The review is short and sweet. Layara et al. provided a good summary of inflammation in aGVHD. I endorse the publication of this review without any further modifications.
We are grateful for this comment.
Round 2
Reviewer 3 Report
The authors appear to have addressed by concerns, although some minor grammatical errors are still and present including new text, which should be detected by the typesetters. One e.g. but there are others is "Purine" at line 194 would read better as "Purines" to match other sub-headings. Another is Figure 2 is cropped.